# Eco-Efficiency Assessment of Japanese Municipalities Based on Environmental Impacts and Gross Regional Product

**Junya Yamasaki [1,]*** , **Toshiharu Ikaga [2]** and **Norihiro Itsubo [3]**

1   Graduate School of Science and Technology, Keio University, 3-14-1 Hiyoshi, Kohokuku, Yokohama, Kanagawa 223-8522, Japan
2   Faculty of Science and Technology, Keio University, 3-14-1 Hiyoshi, Kohokuku, Yokohama, Kanagawa 223-8522, Japan
3   Faculty of Environmental Studies, Tokyo City University, 3-3-1 Ushikubonishi, Tsuzukiku, Yokohama, Kanagawa 224-0015, Japan
*   Correspondence: woodenbat@a6.keio.jp; Tel.: +81-45-566-1770

**Abstract:** Governments at different levels need to appreciate the environmental impacts of socioeconomic activities within their boundaries. They also need to decide relevant environmental policies after carefully examining future pathways based on the relationship between the environment and the economy. This study focuses on Japanese basic administrative divisions (i.e., municipalities) and attempts to quantify the annual environmental efficiency of processes and socioeconomic activities within each of these divisions using life-cycle impact assessment (LCIA) concepts. A key element of the LCIA is the integration of different environmental loads across various impact categories, such as global warming, air pollution, and land use, and their representation through a simple indicator. First, we conduct annual environmental impact assessments for all Japanese municipalities based on reliable, verifiable, and comparable statistical information. Next, we estimate the environmental efficiency of socioeconomic activities within each division by dividing the gross regional product (GRP) with the environmental damage amounts calculated through LIME2, an LCIA-based tool tailored for Japan. Assessment results for each municipality are visualized on maps of Japan in order to highlight the spatial distribution of the values for each indicator. The findings of this study can aid local, regional, and national governments in Japan to inform environmental policy design and decision-making at different spatial levels.

**Keywords:** LCIA method; local government; statistical information; gross regional product; environmental accounting

---

## 1. Introduction

Recent international frameworks such as the Sustainable Development Goals (SDGs) and the Paris Agreement aim at promoting global cooperation toward environmental protection. At the same time, more and more companies have started recognizing the importance of developing and implementing good environmental practices, thus carefully examining their future plans and considering their business activities in relation to the environment and the economy. In this context, the United Nations published in 2012 "The System of Environmental Economic Accounting (SEEA)" [1], an international standard framework to integrate economic and environmental information.

Similarly, many public agencies, including local governments, have also begun incorporating such approaches as part of their financial accounting practices. Indeed, there are many such examples ranging from local to national governments: e.g., (a) the Eurobodalla Shire in Australia publishes annual

statistics on environmental asset values and expenditures, and considers them for environmental conservation within the administrative division [2]; (b) Washington State in the U.S. publishes a strategic plan and a draft budget for a statewide environmental project every two years [3]; and (c) the statistical department of the United Kingdom publishes annual environmental reports for the entire nation based on the SEEA framework [4].

However, compared to private companies and national governments, it may be more difficult for local governments to measure in an objective way the environmental impacts of socioeconomic activity within their own administrative boundaries [5]. This is largely because such assessments must be conducted over wide geographical areas, requiring the expertise of specialists from many different environmental fields [5]. Furthermore, most of the relevant statistical information is often collected and synthesized at the national level, rather than at sub-national levels [5]. However local governments often lack the necessary resources to undertake such assessments, and to date, no country has established a unified system of environmental accounting at the local government level [5]. As a consequence, environmental accounting has not been adopted by local governments as widely as by private companies [5].

In Japan, enterprises started self-reporting their environmental performance in monetary terms in the early 1990s, with many private companies becoming early adopters [5]. The Ministry of the Environment assisted such efforts by publishing its "Environmental Accounting Guidelines 2005" [6], which lays out a system through which organizations can report their environmental activities in monetary terms. The revised version, "Environmental Reporting Guidelines 2018" [7], describes a unified system of environmental principles applicable to all domestic organizations, and seeks to promote environmental accounting more aggressively. These guidelines have been widely adopted and utilized by Japanese companies [5].

Some local governments have also proactively introduced such practices. However, whereas private companies can use the guidelines outlined above, the government of Japan has not provided any official environmental accounting guidelines for public agencies. Hence local governments must devise their own individual methods to assess their local environmental conditions. For example, the cities of Yokosuka and Sabae now publish annual environmental accounting reports using customized methods [8,9].

In reality, however, there is a relative scarcity of studies on environmental accounting by governments, and especially at the local level. For example, an analysis of UK local councils in the 2000s examined the modality of environmental accounting [10]. Other scholars have proposed improvements in the environmental accounting processes undertaken by the government of New South Wales in Australia with respect to waste management [11–13], and the framework for evaluating the economic damages of air pollution from different industries in the U.S. [14]. In addition, some studies have surveyed and analyzed the actual environmental management conditions of specific local governments around the world [15–17]. However, these studies have shown that estimating the environmental loads at the local government level are both limited and usually focus on only single environmental categories. In this respect there is a need to both understand better the environmental performance of local governments, as well as to develop new frameworks that integrate various impact categories and comprehensively assess environmental loads.

Life-cycle impact assessment (LCIA) is a research strand within the field life-cycle assessment literature, which can form the development of such frameworks. A key element of the LCIA is the "integration" under a simple single indicator of different environmental loads that affect various impact categories, such as global warming, air pollution, and land use. Examples of such assessment methods include Extern E [18] and EPS [19]. Additionally, other studies have measured environmental loads at different spatial scales (e.g., countries, regions) using approaches such as carbon and land footprint [20–25]. However, we currently lack methodologies that incorporate these approaches for environmental accounting at the global scale, as well as international standards to assure the quality and enable the comparability of the results [25]. In this context it is therefore possible that LCIA

concepts may form the basis of a unified methodology for environmental accounting at different levels, including at the level of the local government. Such a comprehensive and powerful methodology could, in turn, be used to make informed decisions concerning environmental policies, including at the local level.

This study uses the LCIA method to comprehensively measure environmental loads from basic administrative divisions (i.e., municipalities) in Japan. This study leverages the assessment theory of the LCIA-based method Endpoint Modeling 2 (LIME2) [26,27]. This endpoint-type LCIA method can be used to calculate environmental impacts that reflect environmental conditions and knowledge unique to Japan. LIME2 incorporates the abovementioned "integration" theory of LCIA, and calculates assessment results in monetary units called the "Eco-index Yen" (unit: JPY), while integrating the environmental loads of several impact categories.

Naturally, it is impossible to understand all the processes and socioeconomic activities within each administrative division in the country. Hence the aim of this study is to conduct an environmental accounting of Japanese local governments as comprehensively as possible within the range of LIME2, using statistical information available in Japan. In addition, in this study we conceptualize environmental efficiency, and compare the assessment results with the area, population, and gross regional product (GRP) of each administrative division.

Section 2 outlines the methodology used in this study. Section 3 presents the main results, including a comparative assessment of the environmental efficiencies of Japanese municipalities based on these indicators. Section 4 synthesizes the main findings and identifies some insights to facilitate the use of such integrated techniques to aid public administrators and decision-making with respect to the environment.

## 2. Methodology

### 2.1. Research Approach

This study used LIME2 to assess the environmental loads of administrative divisions (i.e., municipalities) in Japan. The assessment results were divided by the area and population of each municipality to quantify environmental efficiency during a given period. In addition, the environmental efficiency of production activities was estimated for each administrative division by dividing the annual GRP by the annual values of environmental loads. These results were visualized on a map of Japan to illustrate the regional distribution of these environmental loads.

The assessment period was set to one year, based on the assumption that assessment results will correspond to the fiscal years of local governments, as well as the enterprises in their administrative division. This allowed for definitive comparisons of industrial and environmental statistics. For the purposes of this assessment, the year 2015 was chosen because a comparatively large number of relevant statistical information is available for that year.

The scope for assessing environmental loads is defined in accordance with the role of the local government. According to Japanese law, local governments shall autonomously and comprehensively carry out public administration, mainly for the purpose of improving the welfare of local residents. Thus, this study defines the scope of environmental loads, as all operations carried out within the area of the administrative division, and within the range for which the required statistics are available. This is based on the assumption that municipalities have responsibilities that provide them a broad perspective of the current circumstances throughout their divisions. This includes not only public works undertaken by municipalities, but also the operations of private companies and household activity. Incidentally, it excludes household and transport statistics when calculating the environmental efficiency of production activities.

## 2.2. LIME2 Model

This section provides a summary of the LIME2 assessment method [26,27]. LIME2 is an endpoint-type LCIA method developed in Japan and is based on environmental conditions and knowledge unique to Japan. LCIA systems generally comprise two processes, characterization and integration. Characterization is a process for measuring the environmental impacts of products and services throughout their life cycles for a specific impact category. Integration is a process for obtaining an assessment result for a single indicator, by integrating the environmental impacts of several impact categories [26]. The assessment theory of LIME2 includes both of these processes and provides assessment results in terms of the cost of environmental impacts over a certain period using the monetary indicator Eco-index Yen (unit: Japanese yen). The assessment framework of LIME2 is shown in Figure 1.

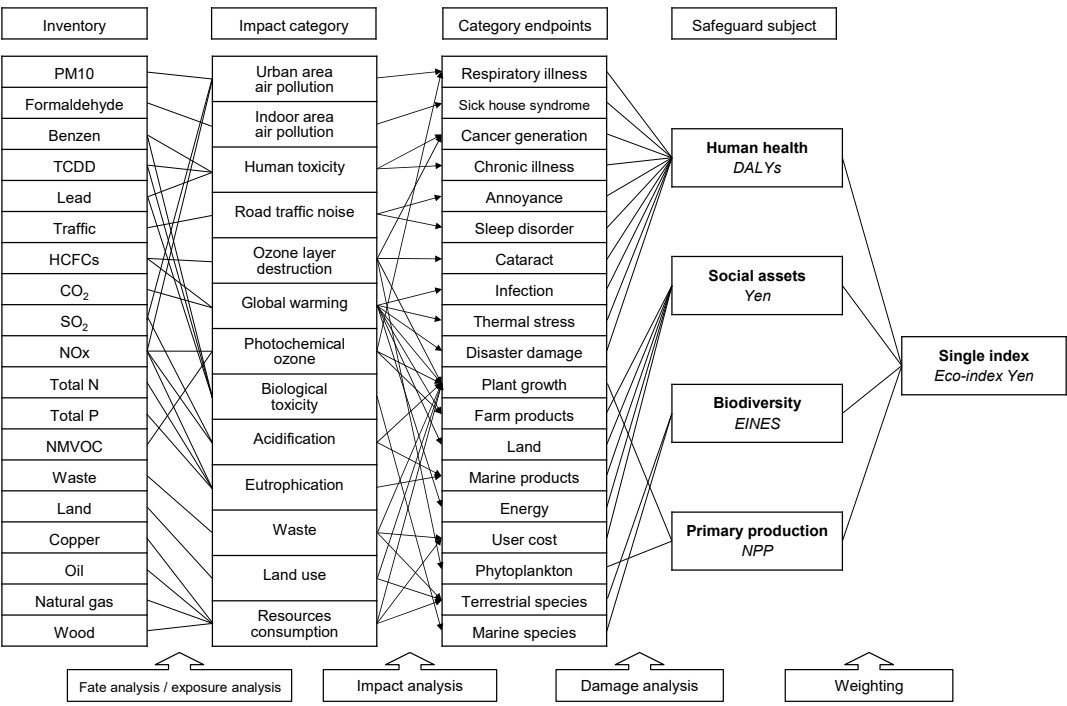

**Figure 1.** LIME2 assessment framework, Source [26].

The LIME2 framework comprises 13 environmental impact categories (e.g., urban air pollution, global warming), with one or more inventories designated for each impact category. The assessment follows three distinct steps as follows. First, each impact category is characterized according to an inventory. Second, damage assessments are conducted to measure the effects of each impact category on each category endpoint (e.g., respiratory illness, disaster damage). Third, for each category endpoint, impact assessments are performed for four safeguard subjects, namely human health, social assets, biodiversity, and primary production. Finally, the results of each impact assessment are integrated into a single indicator.

In LIME2, the stated preference method is used to evaluate the economic impact of environmental loads. The stated preference method is derived from economic science, whereby certain values are elicited through the value judgments of individuals. Stated preference methods are used for measuring abstract values when making comparisons between different subjects. Through a process of integration in LIME2, the assessment results of the four safeguard subjects are weighted through conjoint analysis based on a questionnaire survey of people's values. The assessment results are converted into monetary values, and can be viewed as a reflection of Japanese environmental values. This approach allows for the comparability of environmental loads of several inventories. For all inventories, LIME2 has integration factors that are calculated to reflect each step in the assessment procedure. The single

indicator is directly obtained by multiplying the integration factors with corresponding inventory data and summing these values as follows.

$$SI = \sum_X Inv(X) \times IF(X). \tag{1}$$

*SI*　　　　Single indicator (Eco-index yen) (JPY)
*Inv*(*X*)　　Inventory of substance X (kg)
*IF*(*X*)　　Integration factor of substance X (JPY/kg)

In LIME2, environmental impacts are calculated according to where environmental loads are produced within Japan. The assessment calculation does not change even if a product that is produced in Japan is exported and consumed in another country, as the environmental load resulting from production can have a domestic impact. In contrast, the impact category "resource consumption" takes into account products imported into Japan. LIME2 can assess the environmental impact of consuming certain natural resources. However, because Japan relies heavily on the import of natural resources from other countries, the environmental impact on other countries are calculated according Japan's average annual imports of natural resource.

The integration factors of two impact categories (i.e., photochemical ozone, atmospheric pollution) are provided by Japanese region in order to calculate the environmental impact as a reflection of each environmental condition, such as climate. However, the present study uses the integration factors provided as a standard value for all regions in Japan to assess all impact categories uniformly. This is because one purpose of this study is to assess all administrative divisions based on the same conditions in order to allow for the comparability of the results according to differences in inventory data.

### 2.3. Identifying the Location of Environmental Loads

A major aspect of LIME2 is the localization of environmental loads. To achieve this, it is necessary to standardize the method for estimating the individual transboundary movement of products between divisions. This can allow for the measurement of environmental loads from administrative divisions. However, if environmental loads decline in a certain division that host manufacturing facilities, this outcome should not be interpreted as solely the effort of the municipality. This is because there is also substantial cooperation among industries between administrative divisions.

Several studies have discussed the responsibility over environmental loads for to certain impact categories. For example, inter-industry analyses have been used to examine the sharing of responsibility for greenhouse gas (GHG) emissions between producers and consumers of certain products [28–34]. Other studies have discussed the responsibilities of GHG emissions [35,36] and the shared environmental responsibilities for marine ecosystems [37]. In these studies, scholars basically develop frameworks of environmental responsibility by classifying emitters as either producers (i.e., places of production) or consumers (i.e., places of consumption).

However, it might have proven difficult to apply this framework in the assessment context of this study, in which the environmental loads of various impact categories from various operations were targeted. While it may be possible to draw clear distinctions between producers and consumers of agricultural crops and manufacturing products throughout their life cycles, it is difficult to delineate the environmental responsibility of service industries (e.g., GHG emissions and land use for medical services and education). Accordingly, we adopted a different approach for the assessment reported in this study, which focuses on the environmental responsibilities of administrative units at municipality level.

Thus, this study tentatively proposes two principles for assessing the environmental loads from Japanese divisions: territorial occurrence and territorial benefit. Territorial occurrence is defined as the environmental loads emitted through the life cycles of products or services, which includes the production, consumption (operation), and disposal, and is allocated to the administrative divisions where these loads are emitted. In contrast, territorial benefit is defined as the environmental loads in the

administrative division that benefits from the product or service. In this way, it is possible to achieve a more accurate assessment of the environmental impact of products and/or services. For example, when a product is produced in area A and consumed in area B, the environmental loads associated with the production of the product are allocated to area A under the former principle, and to area B under the latter. Furthermore, when a patient lives in area C and is treated at a hospital in area D, then the environmental loads associated with the treatment are allocated to area D under the former principle and to area C under the latter.

Assessments based on the principle of territorial benefit are better suited to capture the individual effort and responsibility of each municipality. For such assessments, it is necessary to prepare extensive inventory data reflecting the transboundary movement of products and service coverage across all domestic divisions, as well as internationally.

However, it is not easy to accurately track industrial structure at the municipal level in Japan. As a result, in this study we used the principle of territorial occurrence due to its precision with respect to assessment results and the availability of required data. This allowed for the practical calculation of environmental loads in administrative divisions nationwide, whose value appears to be comparable to the GRP for each administrative division. This is because GRP is accounted for in the divisions where the added value is produced.

## 2.4. Assigning Responsibility for Environmental Loads among Municipalities

As LIME2 reports results as a monetary indicator, it is important assess the responsibility of different municipalities for environmental loads. Even though LIME2 is utilized in Japan, the areas affected by environmental impacts are much broader (i.e., regions within Japan, Japan as a whole, entire world), and differ between impact categories. For example, the impact area of respiratory illness caused by air pollutants is limited to a certain region, i.e., region where these particles are suspended in the atmosphere. In contrast, the impact area related to the global warming caused by GHG emissions is the entire world.

The assessment theory of LIME2 is based on individual impact areas according to each impact category (Section 2.1). Therefore, the assessment results based on this method include environmental loads that affect foreign countries (either partly or completely). Hence it may be excessive for Japanese municipalities to interpret these environmental loads as their individual responsibility. It is thus necessary to arrange these concepts in order to apply this method to the municipality level.

This study takes the position that the Japanese government has complete responsibility for the environmental loads emitted within Japanese territory. We base this decision based on the fact that Japan is a member of several international organizations, and has been recognized as a nation that has both a great responsibility for the global environment, and has exerted leadership towards this end.

Accordingly, this study accounts for all environmental loads calculated in Japan toward the total damage on environmental assets both at home and abroad, with this responsibility belonging to the Japanese government. Here, environmental assets are interpreted as the four safeguard subjects mentioned in Section 2.1, namely human health, social assets, biodiversity, and primary production. In this study, the assessment results expressed in a monetary unit is called the "damage amount on environmental assets". Based on this interpretation, Japanese municipalities shall uniformly be responsible for their individual damage amounts as they are part of the Japanese nation.

## 2.5. Data Inventory

One of the most resource and time consuming stages of this assessment is the preparation of LIME2 inventory data from national and local statistical sources. The LIME2 assessment framework comprises 13 impact categories (Section 2.1), each of which is made up of several inventory items. However, because it is not practical to prepare comprehensively all this inventory data, it is necessary to select which items to include based on the assessment purpose, information availability, and data

accuracy. Thus, we surveyed before starting the study the availability of statistical information for all inventory items at the level of Japanese municipalities.

We used publicly available statistical information that was uniformly collected or estimated by governmental agencies in Japan to ensure that the inventory data is reliable, verifiable, and comparable across all divisions in order to allow for a valid assessment of Japanese municipalities nationwide. Specifically, we used statistical information from the year 2015. When data was not available for the year 2015, we used data from the year closest to 2015.

Japan consists of 47 prefectures, which in turn consist of 1747 municipalities. This study prioritizes the uniform use of statistical information across all Japanese municipalities, aiming to provide useful knowledge to local governments throughout the country. When statistical information at the municipal level is unavailable, we generated estimates based on information collected at the prefectural level. In the end, the statistical information of six municipalities (i.e., Shikotan, Tomari, Ruyobetsu, Rubetsu, Shana, and Shibetoro in Hokkaido) was insufficient and therefore removed from the final assessments.

Table 1 describes, for all 13 impact categories, the inventory items for which data was available, the number of inventory items, the government ministry that reported the statistical information, the assessment indicator, and the minimum administrative unit for which data was available.

**Table 1.** LIME2 inventory data for Japanese administrative divisions.

| Impact Category | Inventory (Number) | Government Agency | Indicator | Administrative Unit |
|---|---|---|---|---|
| Ozone layer destruction | Various chemicals (21) | Ministry of Economy, Trade and Industry [38] | Emission amount [kg] | Municipalities |
| Photochemical ozone | Various chemicals (58) | | | |
| Human toxicity | Various chemicals (99) | | | |
| Biological toxicity | Various chemicals (127) | | | |
| Eutrophication | Various chemicals (17) | | | |
| Global warming | Various chemicals (19) | | | |
| | $CO_2$ (1) | Ministry of the Environment [39] | Emission amount [kg] | Municipalities |
| Land use | Various types of land use (5) | Ministry of Land, Infrastructure, Transport and Tourism [40] | Current area [km$^2$] | Municipalities |
| Resource consumption | Coal, natural gas, crude oil (3) | Ministry of Economy, Trade and Industry [41] | Consumption amount [kg] | Prefectures |
| Acidification | SOx, NOx (2) | Ministry of the Environment [42] | Emission amount [kg] | Prefectures, and some municipalities |
| Atmospheric pollution | SOx, NOx (2) | | | |
| Waste | Various types of domestic waste (1) | Ministry of the Environment [43] | Disposal amount [kg] | Municipalities |
| Road traffic noise | Travel distances by type of car (4) | Ministry of Land, Infrastructure, Transport and Tourism [44] | Travel distance [km] | Prefectures, and some municipalities |
| Indoor air pollution | Various chemicals (-) | - | - | - |

Note: The numbers in parentheses indicate the number of inventory items for which data is available.

As shown in the rightmost column of Table 1, some inventory data is not available at the municipal level for the four impact categories: resource consumption, acidification, urban air pollution, and road traffic noise. As previously mentioned, we estimated municipal loads from the prefectural data, using a method published by the Japanese Ministry of the Environment [45]. As shown in the sixth row of Table 1, the Ministry of the Environment publishes statistical information about annual $CO_2$ emissions at the municipal level. In this study, the $CO_2$ emission data at the municipal level was estimated from the data at the prefectural level, based on proportion distribution using other indicators as follows.

$$Inv_{mun}(X) = Inv_{pre}(X) \times \frac{D_{mun}}{D_{pre}}. \tag{2}$$

| $Inv_{mun}$ (X) | Inventory data at the municipal level |
| --- | --- |
| $Inv_{pre}$ (X) | Inventory data at the prefectural level |
| $D_{mun}$ | Other indicator for proportion distribution at the municipal level |
| $D_{pre}$ | Other indicator for proportion distribution at the prefectural level |

In its method, the Ministry of the Environment uses as the factors for "D" in Equation (2) the indicators "shipment value of manufactured goods" in the manufacturing sector, "number of employees" in industrial sectors except manufacturing, "number of households" in the residential sector, and "number of automobiles owned in divisions" in the transportation sector. The Ministry explains that this is one of the simplest methods to estimate data at the municipal level from data at the prefectural level, and it is suitable for approximating the overall data for each administrative division in Japan [45].

In the global warming category, the Ministry of the Environment collects data for six types of GHG emissions: $CO_2$, $CH_4$, $N_2O$, HFCs (hydrofluorocarbons), PFCs (perfluorocarbons), and $SF_6$. The Ministry converts the greenhouse effect of these substances into the equivalent effect of $CO_2$, and expresses it in terms of the equivalent mass of $CO_2$. Accordingly, this study refers to this data as $CO_2$ emission and uniformly uses the integration factor of $CO_2$. Incidentally, the 19 substances accounted for global warming by the Ministry of Economy, Trade and Industry do not include these six types of GHGs.

The data on land use comes from geographic information system (GIS) data published by the Ministry of Land, Infrastructure, Transport, and Tourism. This data shows different types of land use, indicated as percentage per $km^2$. For this study, we converted the data for the administrative divisions of each municipality using ArcGIS (v 10.5) software. Assessment targets are limited to man-made land use types, such as paddy fields, croplands, buildings, roads, and other sites (e.g., golf courses).

LIME2 assesses waste impacts according to waste type, including paper, plastic, metal, and so on. However, the required data about disposal conditions according to type published in Japanese statistical information is incomplete. Therefore, for this study we used the integration factor prepared as a standard for all types of waste for the total disposal amount of all types of waste by each municipality.

For indoor air pollution, LIME2 assesses environmental loads based on the mass of toxic chemical emissions in residential houses during construction and use. While this inventory data is required for the assessment, it is not easy to collect or estimate it for all residential houses in each administrative division. Moreover, this category does not correspond with the concept of assessing all municipalities nationwide, and was thus excluded from the assessment.

## 2.6. Conceptualizing Environmental Efficiency

The environmental loads of individual municipalities are thought to be largely related to area and population size, with larger municipalities tending to produce larger environmental loads. Accordingly, this study attempts at conceptualizing environmental efficiency by dividing the values derived from the assessment results by administrative area and population in 2015. This makes it possible to compare environmental loads separately from size and to examine the qualities of each municipality from various perspectives.

However, the damage amount per unit area and per capita is thought to be higher in administrative divisions with more active industries. Therefore, it is not appropriate to directly assess the environmental efficiency of municipalities based solely on these indicators. In order to measure eco-efficiency more accurately, it is necessary to account for the benefits realized from these processes resulting in these damage amounts. This study therefore attempts to assess the environmental efficiency of administrative divisions by using GRP as a factor of analysis (the GRP denotes the total value added by all industries in a given division over a given period in monetary terms. The Gross Domestic Product (GDP) is commonly used at the national level, while the GRP at the regional level within a country).

The GRP is one of the most representative indicators for this purpose, and the statistical information is readily available for each Japanese municipality. Accordingly, in this study we define the unique index by dividing the value of GRP (unit: Japanese yen) by the value of damage amount (unit: Japanese yen, JPY). This indicator is called the "environmental efficiency (unit: dimensionless)", and is calculated as follows.

$$(Environmental\ efficiency)\ [-] = \frac{(GRP)\ [\text{JPY}]}{(Damage\ amount)\ [\text{JPY}]}. \tag{3}$$

This study uses to the statistical information each municipality's GRP as published by the Ministry of Economy, Trade, and Industry. As this statistical analysis was not conducted in 2015, we instead used the data from 2016. Furthermore, we used the indicator for normal GRP, which is based on commodity prices in 2016. Furthermore, it is also necessary to recalculate the damage amounts for certain sectors assessed through GRP indicators in order to make them correspond to both indicators for the assessment. Thus, the damage amount is limited to the amount corresponding to the industry sector (which includes the service industry) in all impact categories (i.e., damage amounts for the household and transport sectors were excluded from this analysis). This indicator of the final damage amount is called the "production damage amount" in this study. For example, the $CO_2$ emissions related to the consumption of heating energy in households was not included in the damage amount for the global warming category. Additionally, the damage amounts for domestic waste and road traffic noise were excluded as these damages are wholly caused by the household and transport sectors (see above). We also reluctantly excluded the damage amounts for land use because it is impossible to classify these damages for certain sectors (e.g., building site for industrial sector and household sector) due to the nature of the statistical information. Incidentally, it is preferable to use damage amounts under the principle of territorial occurrence because the administrative division where the added value is accounted for, is the same division where the related environmental load is emitted.

Finally, we should point that in economic terms, a "flow" denotes the amounts that something changes over a given period of time, and "stock" the storage of something at a particular point in time [46]. The GRP and production damage amount both measure the flow within administrative divisions. Thus, environmental efficiency can be interpreted as an index of the relationship between the amounts of change for both indicators over a given period.

## 3. Results

Sections 3.1–3.5 present the assessment results for Japanese administrative divisions (i.e., municipalities) as calculated through the abovementioned methods (Section 2). The environmental load calculated by LIME2 is called the damage amount on environmental assets, as previously mentioned in the Section 2.4. Though these calculations were originally performed in Japanese yen (JPY), the results are presented here in U.S. dollars (111.34 USD/JPY on 1 April 2019). For reference, the tables and figures showing the results in Japanese yen are included in the Supplementary materials.

### 3.1. Total Damages for All of Japan

Total damage amounts for all of Japan are shown in Figure 2 (the figure in JPY is in the Supplementary materials, Figure S1). The total damage amount in 2015 was USD 76.6 billion (JPY 8.53 trillion). The impact category with the largest amount was global warming (USD 25.3 billion, JPY 2.82 trillion), followed by land use (USD 20.9 billion, JPY 2.33 trillion), domestic waste (USD 11.1 billion, JPY 1.24 trillion), biological toxicity (USD 5.66 billion, JPY 0.63 trillion), atmospheric pollution (USD 5.48 billion, JPY 0.61 trillion), and resource consumption (USD 5.03 billion, JPY 0.56 trillion). These six categories accounted for 96.1% of the total damages.

Figure 3 provides a breakdown of inventory data for the categories according to the total damage amount (the figure in JPY is in the Supplementary materials, Figure S2). As it is, some specific indicators dominate the specific categories. For example, building sites accounted for 61.0% of the total damages associated with land use. Likewise, styrene accounted for 91.9% of the total biological toxicity costs,

SOx accounted for 88.8% for total atmospheric pollution costs, and crude oil accounted for 78.4% of the total resource consumption costs. Incidentally, nearly all of the global warming costs consisted of the inventory data collected by the Ministry of the Environment on six GHG types, including $CO_2$. Moreover, the entire domestic waste cost was accounted for by a single inventory dataset, as previously mentioned.

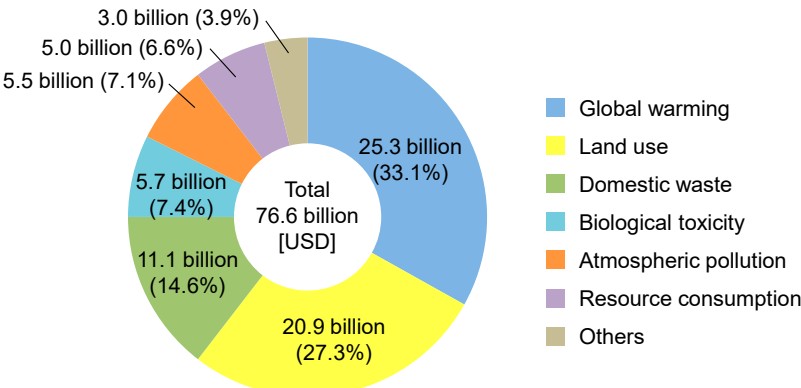

**Figure 2.** Total damage amounts for Japan by impact category (in USD).

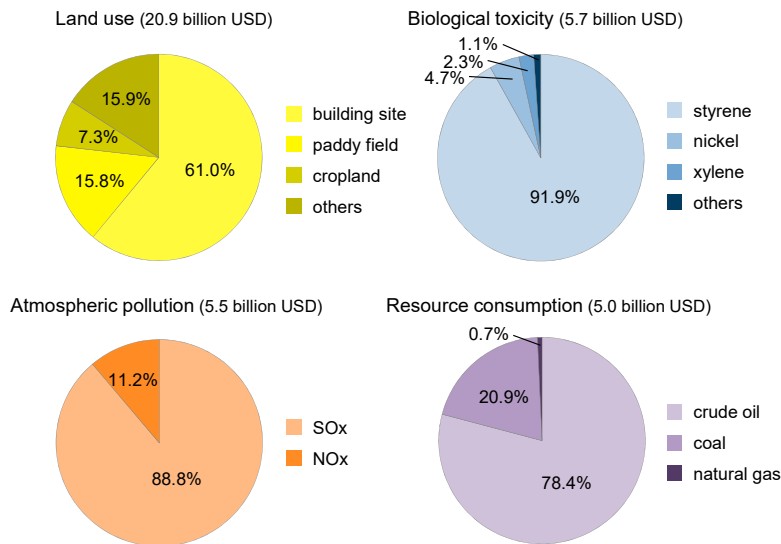

**Figure 3.** Breakdowns of inventory data for total damage amount by major impact categories (in USD).

### 3.2. Disaggregated Damages for All of Japan

Table 2 provides the damage amounts per unit area and per capita for all municipalities, including lists of the average amounts, standard deviation, and variation coefficient for each impact category (the table in JPY is in the Supplementary materials, Table S1).

The averages for the total damage amount per unit area and per capita are calculated at 666,000 USD/km$^2$ (74.2 million JPY/km$^2$) and 916 USD/capita (102,000 JPY/capita). The top three damage amounts per unit area are global warming (186,000 USD/km$^2$, 20.7 million JPY/km$^2$), land use (130,000 USD/km$^2$, 14.5 million JPY/km$^2$), and domestic waste (70,200 USD/km$^2$, 7.82 million JPY/km$^2$). The top three damage amounts per capita are land use (419 USD/capita, 46,700 JPY/capita), global warming (231 USD/capita, 25,700 JPY/capita), and domestic waste (86.3 USD/capita, 9610 JPY/capita).

The wide range of differences between the average damage amounts per unit area and per capita is due to the wide range of the land area and population of municipalities throughout Japan. The variation coefficients for ozone layer destruction, photochemical ozone, human toxicity, biological toxicity, and eutrophication are much higher than the total values, suggesting that the damage amounts

for these categories vary substantially between municipalities. The inventories of these impact categories include the toxic chemical substances produced continuously from specific industries, such as the heavy chemical industry. Therefore, there are major disparities in damage estimates between municipalities according to the types of industries contained within their divisions.

**Table 2.** Damage amounts per unit area and per capita.

| Impact Category | Damage Amount per Unit Area | | | Damage Amount per Capita | | |
|---|---|---|---|---|---|---|
| | Average (USD/km$^2$) | Standard Deviation (USD/km$^2$) | Variation Coefficient (-) | Average (USD/Capita) | Standard Deviation (USD/Capita) | Variation Coefficient (-) |
| Ozone layer destruction | $1.27 \times 10^3$ | $3.52 \times 10^4$ | 27.91 | $8.30 \times 10^{-1}$ | $2.06 \times 10^1$ | 24.78 |
| Photochemical ozone | $2.06 \times 10^3$ | $1.71 \times 10^4$ | 8.28 | $2.87 \times 10^0$ | $2.61 \times 10^1$ | 9.12 |
| Human toxicity | $3.85 \times 10^3$ | $4.20 \times 10^4$ | 10.91 | $2.60 \times 10^0$ | $2.26 \times 10^1$ | 8.70 |
| Biological toxicity | $6.06 \times 10^4$ | $3.60 \times 10^5$ | 5.94 | $5.46 \times 10^1$ | $5.14 \times 10^2$ | 9.41 |
| Eutrophication | $1.36 \times 10^{-3}$ | $2.88 \times 10^{-2}$ | 21.23 | $1.97 \times 10^{-6}$ | $4.89 \times 10^{-5}$ | 24.81 |
| Global warming | $1.86 \times 10^5$ | $4.82 \times 10^5$ | 2.59 | $2.31 \times 10^2$ | $3.06 \times 10^2$ | 1.33 |
| Land use | $1.30 \times 10^5$ | $1.31 \times 10^5$ | 1.01 | $4.19 \times 10^2$ | $5.27 \times 10^2$ | 1.26 |
| Resource consumption | $3.51 \times 10^4$ | $1.01 \times 10^5$ | 2.86 | $3.69 \times 10^1$ | $1.01 \times 10^2$ | 2.72 |
| Acidification | $3.87 \times 10^3$ | $8.84 \times 10^3$ | 2.28 | $6.75 \times 10^0$ | $7.03 \times 10^0$ | 1.04 |
| Atmospheric pollution | $5.92 \times 10^4$ | $2.01 \times 10^5$ | 3.40 | $4.75 \times 10^1$ | $6.37 \times 10^1$ | 1.34 |
| Domestic waste | $7.02 \times 10^4$ | $1.36 \times 10^5$ | 1.97 | $8.63 \times 10^1$ | $3.14 \times 10^1$ | 0.36 |
| Road traffic noise | $1.07 \times 10^4$ | $1.54 \times 10^4$ | 1.44 | $3.50 \times 10^1$ | $3.85 \times 10^1$ | 1.10 |
| Total | $6.66 \times 10^5$ | $1.61 \times 10^6$ | 2.41 | $9.16 \times 10^2$ | $1.92 \times 10^3$ | 2.10 |

Note: Number of municipalities: 1741. The variation coefficient is calculated by dividing the standard deviation by the average, and it shows the relative variation in the data.

### 3.3. Spatially-Explicit Damages for Japanese Municipalities

In this section, we visualize the assessment results for all municipalities calculated in Section 3.2 on a map of Japan in order to understand the spatial distribution of environmental loads. Figure 4 contains the total damage amount per unit area and per capita for Japanese municipalities for the entire country (the figure in JPY is in the Supplementary materials, Figure S3). Colors represent 10% increments in damage amounts across all municipalities.

The damage amount per unit area tends to be higher in more densely populated areas, and particularly around the three major metropolitan areas of Japan, including the Kanto area (around Tokyo and Yokohama), the Kinki region (around Osaka), and the Chukyo region (around Nagoya). The rate was also relatively high in and around other major cities such as Fukuoka and Sapporo. In contrast, the rate was lower in sparsely populated areas, such as inland mountainous regions and Hokkaido. This distribution is extremely similar to the distribution of population density, indicating a deep relationship between these indicators.

In contrast, the damage amount per capita tends to be lower in densely populated areas, such as the three major metropolitan areas mentioned above. At the same time, it tends to be higher in sparsely populated areas, such as inland mountainous regions. In some urban areas, the absolute damage amount is higher, but the population tends to be concentrated in rates higher than that of the damage amount. It therefore may be suggested that environmental efficiency is comparatively higher around urban areas from the perspective of environmental loads per capita. However, agricultural crops and manufactured goods are produced outside of urban areas, but are consumed within these urban areas. To accurately verify the possibly higher damage amounts accruing from the import of crops/goods in cities it would be essential to develop assessment theories and tools based on the principal of territorial benefit mentioned above (Section 2.3).

The assessment results for the top five impact categories (Figure 2) are visualized in terms of damage amounts per unit area and per capita in Figure 5. The distributions of damage amount per unit area are very similar for four impact categories, namely global warming, land use, domestic waste, and atmospheric pollution. The spatially-explicit damage amounts for these categories tend to be higher in urban and suburban areas because the emissions of GHGs and atmospheric pollutants are

comparatively larger, from not only the industrial sector but also the service and transport sectors. In this respect, these damage amounts tend to be high in densely populated areas, regardless of the type of industries located within these areas. Naturally, the damage amounts for land use and domestic waste also tend to be higher in densely populated areas. In contrast, the damage amounts for biological toxicity is the only category to not show a relationship with population distribution in Figure 5. The distribution of the damage amount for this category is not particularly related to the population because it is largely affected by the type of industry located in a given area, regardless of the area's population.

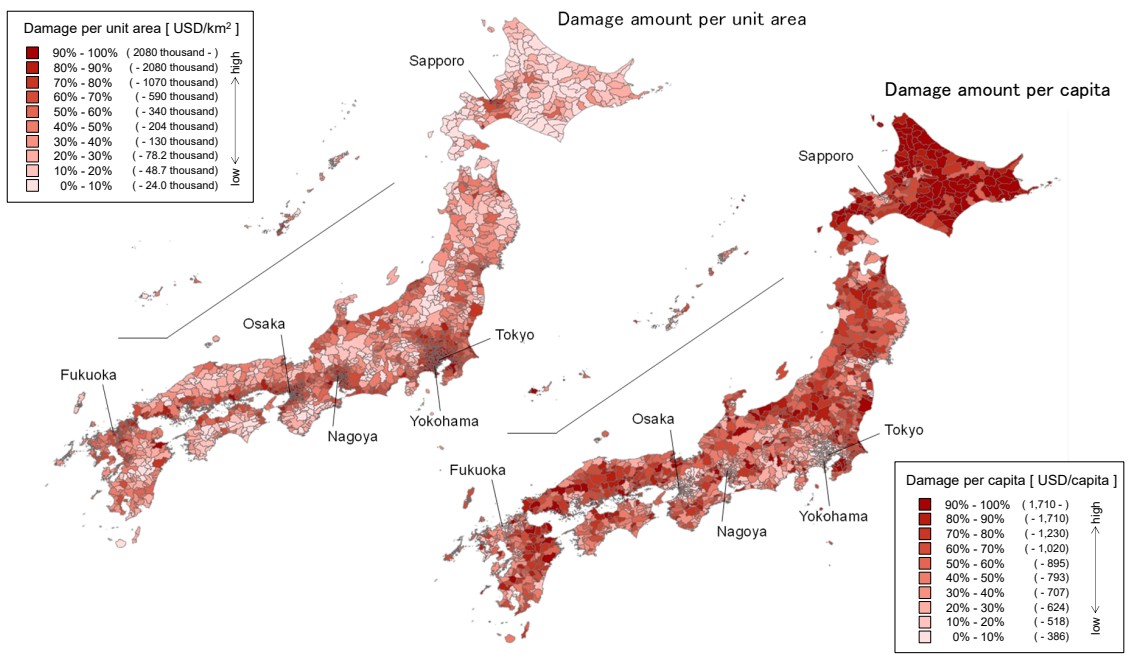

**Figure 4.** Spatially explicit total damage amount in Japanese municipalities (left map, damage amount per unit area; right map, damage amount per capita).

Certain impact categories exhibit different patterns between damage amount per capita, and per unit area. In particular, the damage amounts for global warming, land use, and atmospheric pollution are lower in the central parts of the three major metropolitan areas. This is reflected in the results due to the fact that the populations are larger than the absolute damage amount for each category in these areas, as described above. However, there are some exceptions in the suburbs of these areas. The damage amounts for the global warming and atmospheric pollution being higher in parts of the areas surrounding the national capital region and the Kinki region. These areas contain some industries that actively support the local economies, but the population is not as concentrated as in central parts of the metropolitan regions, as reflected in the results.

Taking a closer look at the damage amount per capita for each impact category, the rate for global warming tends to be higher in cold areas, such as Hokkaido (the northern island that includes Sapporo city), in the coastal areas of western Japan, and in some suburban areas, as described above. In Hokkaido, heating energy consumption is higher in colder areas, which affects the results of the household sector. Heavy industry has been active in the coastal areas of western Japan for a long time. The distribution of atmospheric pollution damages also reflects the distribution of industrial zones in the coastal areas throughout the country. The damage rate for land use is much lower in densely populated areas, in contrast with the amount per unit area. This suggests that environmental efficiency may be higher in more densely populated areas, particularly from the perspective of residential land use. The damage amounts for domestic waste shows a different distribution for amount per capita than per unit area, but it does not show a particular relationship with population distribution. As shown

in the rightmost column of Table 2, the variation coefficient of this impact category per capita is the lowest among all categories, indicating that the disparity for this impact category is comparatively narrow across all municipalities across Japan. Biological toxicity is the only category for which damage amounts show a similar distribution between the amounts per unit area, and amounts per capita. As this category is affected by the specific industries in a given area, both distributions are independent from other categories.

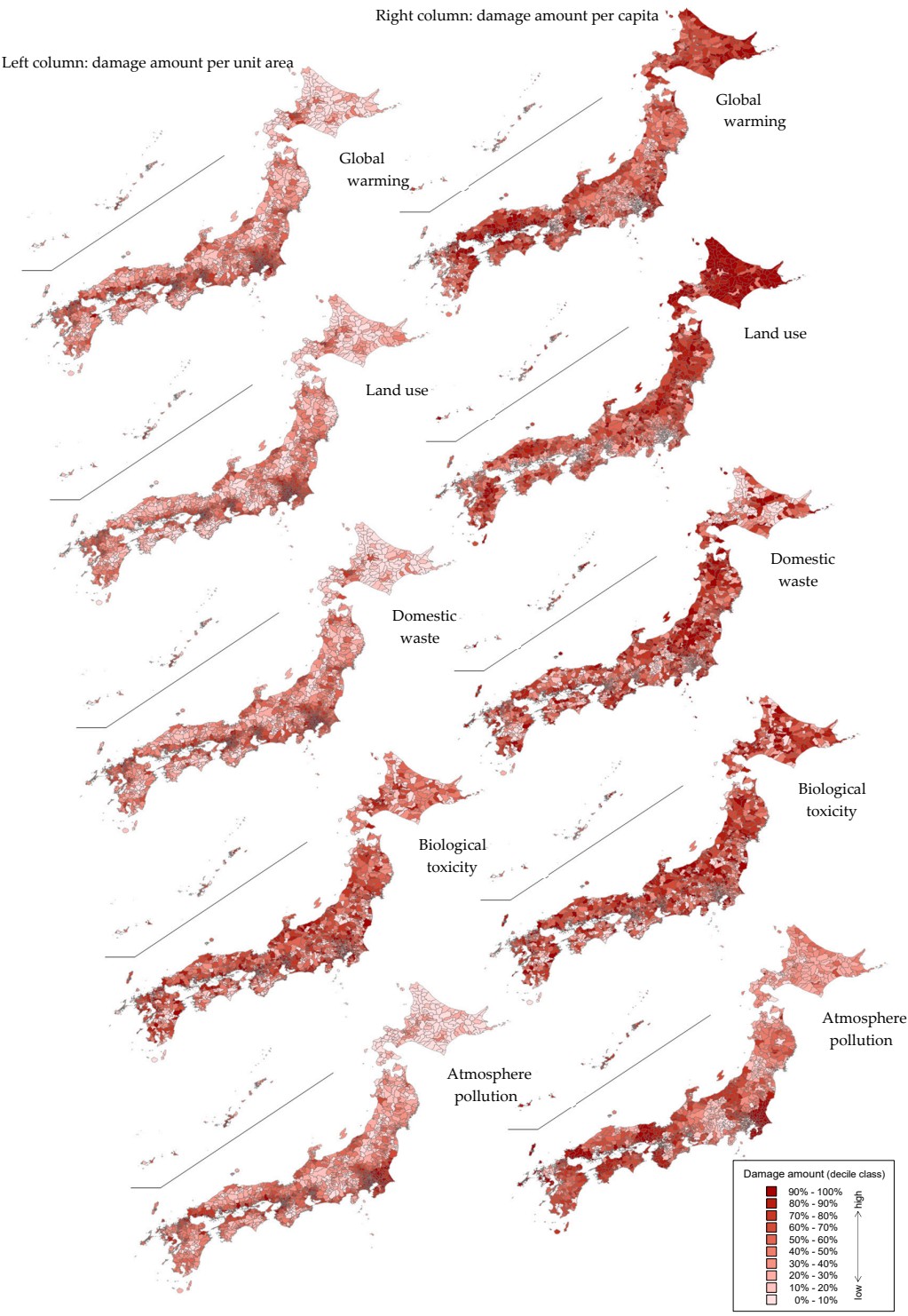

**Figure 5.** Assessment results for each impact category for Japanese municipalities; (left column, damage amount per unit area; right column, damage amount per capita).

### 3.4. Spatially-Explicit Environmental Efficiency for Japanese Municipalities

Figure 6 shows the values of GRP per capita and environmental efficiency (Section 2.6) for all municipalities (the figure in JPY is in the Supplementary materials, Figure S4), in 10% increments. Here, the median value of environmental efficiency for all municipalities is calculated as 65.5. Accordingly, values higher than the median are represented by cold colors (blue) and lower values are represented by warm colors (red) for each administrative division in Figure 6.

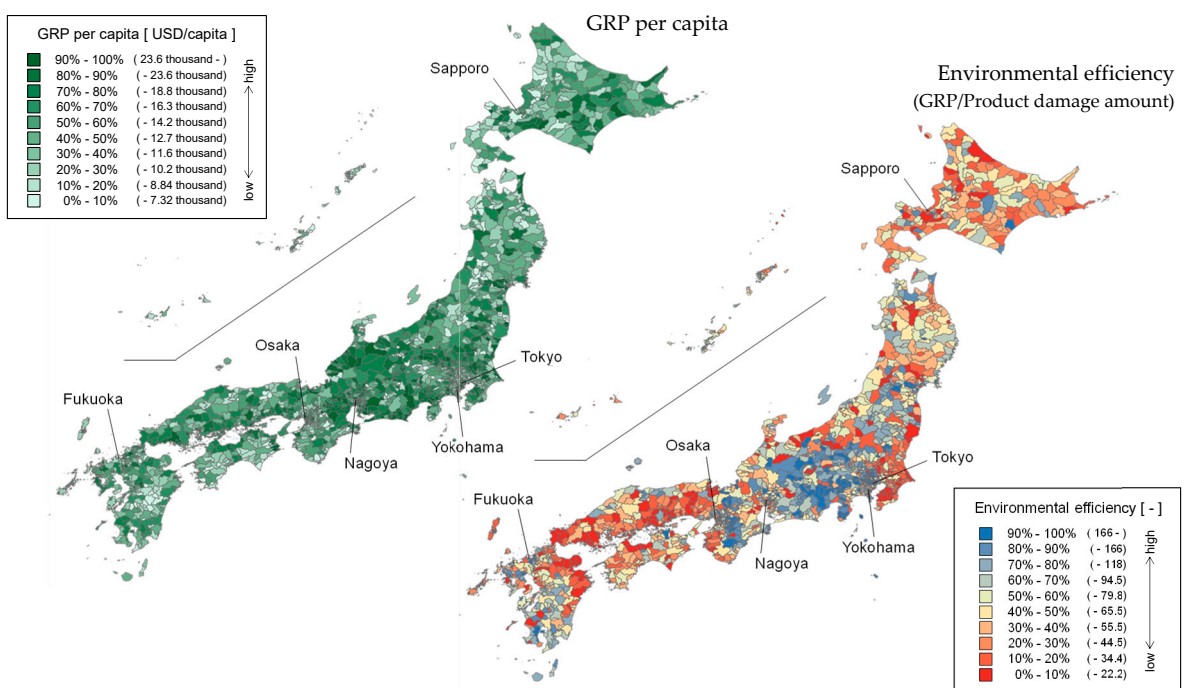

**Figure 6.** Gross regional product (GRP) per capita and environmental efficiency for Japanese municipalities; (left map, GRP per capita; right map, environmental efficiency).

The GRP per capita tends to be higher in the Chubu region (the central part of Japan that includes Nagoya) due to the concentration of the automotive and electronics industries in the region, which added a large value. Moreover, the rate tends to be higher in certain coastal areas around Tokyo and western Japan where heavy industries are concentrated.

As a result, environmental efficiency also tends to be higher in the Chubu region. This implies that the automotive industry produces comparatively larger benefits relative to its environmental loads. Similarly, the benefits produced by the electronics industry are large relative to the unit size of its products, so the value added is larger than the environmental loads. In contrast, the rate tends to be comparatively lower in certain coastal areas around Tokyo and western Japan, as the GRP per capita is higher but the production damage amount per capita is also higher. Heavy industries produce large benefits, but their environmental loads are also large.

### 3.5. Assessment Results for Major Cities

This section focuses on the 21 major urban municipalities of Japan (in Tokyo, the capital of Japan, the main area where the national administrative functions are carried out is designated as a special ward. In this section this area is treated as an administrative division for the municipality of Tokyo. In addition, the 20 municipalities whose populations and economic scales are large are offered a higher degree of autonomy than most other municipalities in Japan. These municipalities are officially called government-designated cities. Table 3 outlines the assessment results for these 21 municipalities), and discusses their environmental performance based on the results outlined in the previous sections. Table 3 includes the assessment results for GRP per capita, production damage amount per capita,

and environmental efficiency, along with the cumulative relative frequency (cumulative relative frequency is an index that shows a position relative to all data in a constellation. The highest value for all municipalities is 100% and the bottom value is 0% for each indicator) for these indicators (the table in JPY is in the Supplementary materials, Table S2). Figure 7 shows the value of production damage amount per capita on the horizontal axis, and the value of GRP per capita on the vertical axis (the figure in JPY is in the Supplementary materials, Figure S5). Each point represents a municipality, with the slope of the line connecting the plot point with origin point showing the value of environmental efficiency for each municipality. Accordingly, the closer to upper left corner of the graph a point is, the higher the environmental efficiency of the municipality.

**Table 3.** Assessment results for 21 major municipalities in Japan.

| Municipality | GRP per Capita (USD/Capita) | | Production Damage Amount per Capita (USD/Capita) | | Environmental Efficiency (-) | |
|---|---|---|---|---|---|---|
| Sapporo | $1.80 \times 10^4$ | (78.8%) | $2.68 \times 10^2$ | (65.6%) | 67.2 | (53.1%) |
| Sendai | $2.66 \times 10^4$ | (94.8%) | $1.86 \times 10^2$ | (47.8%) | 143.1 | (89.1%) |
| Saitama | $1.94 \times 10^4$ | (83.8%) | $5.93 \times 10^1$ | (4.8%) | 327.0 | (99.3%) |
| Chiba | $2.15 \times 10^4$ | (89.1%) | $5.24 \times 10^2$ | (84.9%) | 41.1 | (27.4%) |
| Tokyo (special ward) | $5.28 \times 10^4$ | (99.1%) | $1.48 \times 10^2$ | (38.5%) | 355.0 | (99.6%) |
| Yokohama | $1.92 \times 10^4$ | (83.2%) | $2.20 \times 10^2$ | (56.9%) | 87.3 | (68.0%) |
| Kawasaki | $1.70 \times 10^4$ | (75.0%) | $4.37 \times 10^2$ | (80.5%) | 38.9 | (25.2%) |
| Sagamihara | $1.37 \times 10^4$ | (57.2%) | $1.39 \times 10^2$ | (34.3%) | 98.0 | (74.7%) |
| Niigata | $1.86 \times 10^4$ | (81.2%) | $2.06 \times 10^2$ | (53.3%) | 90.5 | (70.4%) |
| Shizuoka | $2.24 \times 10^4$ | (90.4%) | $1.04 \times 10^2$ | (21.9%) | 214.8 | (96.6%) |
| Hamamatsu | $2.08 \times 10^4$ | (87.4%) | $1.14 \times 10^2$ | (25.8%) | 182.2 | (94.1%) |
| Nagoya | $3.22 \times 10^4$ | (97.0%) | $3.39 \times 10^2$ | (73.5%) | 95.0 | (72.8%) |
| Kyoto | $1.99 \times 10^4$ | (85.5%) | $2.65 \times 10^2$ | (65.3%) | 75.3 | (59.0%) |
| Osaka | $4.62 \times 10^4$ | (98.7%) | $5.06 \times 10^2$ | (84.0%) | 91.4 | (71.0%) |
| Sakai | $1.62 \times 10^4$ | (71.4%) | $2.69 \times 10^2$ | (65.7%) | 60.4 | (45.5%) |
| Kobe | $2.15 \times 10^4$ | (89.1%) | $1.95 \times 10^2$ | (49.9%) | 110.6 | (80.7%) |
| Okayama | $1.95 \times 10^4$ | (83.9%) | $3.04 \times 10^2$ | (70.0%) | 64.0 | (49.3%) |
| Hiroshima | $2.26 \times 10^4$ | (90.9%) | $3.13 \times 10^2$ | (71.2%) | 72.4 | (57.0%) |
| Kitakyushu | $1.82 \times 10^4$ | (79.5%) | $4.43 \times 10^2$ | (80.9%) | 41.1 | (27.4%) |
| Fukuoka | $2.69 \times 10^4$ | (94.9%) | $8.49 \times 10^1$ | (12.8%) | 316.9 | (99.2%) |
| Kumamoto | $1.57 \times 10^4$ | (68.9%) | $1.90 \times 10^2$ | (48.6%) | 82.9 | (64.9%) |

Note: Cumulative relative frequency based on all municipalities is shown in parentheses.

As shown in Table 3 and Figure 7, the top three municipalities in terms of GRP per capita are Tokyo (52,800 USD/capita, 5.88 million JPY/capita), Osaka (46,200 USD/capita, 5.14 million JPY/capita), and Nagoya (32,200 USD/capita, 3.58 million JPY/capita). These municipalities are essentially the centers of the three major metropolitan areas in Japan. The results reflect the thriving economic conditions in these municipalities. Additionally, 15 of the 21 municipalities have a cumulative relative frequency of 80% or greater. This meant that these municipalities are in the top 20% of all Japanese municipalities, and that their productivity is comparatively higher than most other municipalities.

The top three municipalities in terms of production damage amount per capita are Chiba (524 USD/capita, 58,300 JPY/capita), Osaka (506 USD/capita, 56,300 JPY/capita), and Kitakyushu (443 USD/capita, 49,300 JPY/capita). Chiba is located east of Tokyo in the national capital region, and Kitakyushu is located north of Fukuoka in Kyushu island. There are various industries supporting the local economy in these large urban areas, and hence their environmental loads are comparatively higher.

The top three municipalities in terms of environmental efficiency are Tokyo (355.0), Saitama (327.0), and Fukuoka (316.9). In Tokyo, GRP per capita is high but the production damage amount per capita is comparatively low, so it has the highest environmental efficiency among the 21 municipalities. Tokyo is

also a cultural hub, and has many large commercial facilities, entertainment facilities, and major media companies, such as broadcasters and publishing firms. The environmental loads of these businesses are low relative to their benefits, which is one of the reasons for Tokyo's assessment results. Similarly, in Saitama and Fukuoka the GRP per capita is high and the production damage amount per capita is low. The main industries in these municipalities are service businesses, and the assessment results seem to reflect this situation. Additionally, seven of the 21 municipalities have cumulative relative frequency of over 80%, and 16 municipalities of over 50%.

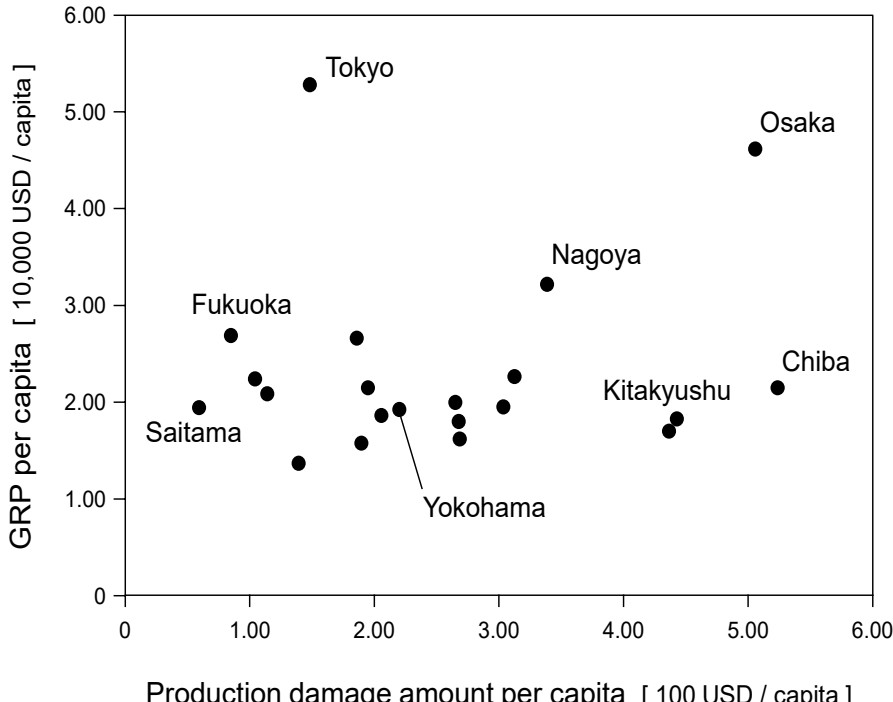

**Figure 7.** Assessment results for 21 major municipalities in Japan.

## 4. Discussion

### 4.1. Practical Application for Local Governments

Local governments can use the assessment theory of LIME2 (and derived tools) for environmental accounting and tracking their environmental performance. In particular, the ability of LIME2 to assess environmental loads across administrative divisions using a single monetary indicator can provide public administrators at different levels, with valuable information. For example, this information can be used to compare the effects of different socioeconomic activities, policies, and interventions across different environmental impacts. Such information could be useful for allocating environmental conservation funds to local governmental budgets. Moreover, such systems could provide urban residents with easy-to-understand information about their local environmental conditions and performance, thanks to the presentation of assessment results using a unified and easily understandable indicator. Such advantages could solve some of the constraints that local governments experience when undertaking environmental accounting (Section 1) [5].

The assessment performed in this study was based on statistical information for the year 2015. If the same assessment is conducted over multiple years, it would be possible to understand changes in environmental damages over time. The degree of change can be interpreted as a proxy for the environmental conservation efforts exerted by local governments. This could allow an estimate of whether environmental policies are successful in reducing environmental loads from municipalities. Similarly, by comparing changes in damage amounts with the costs of environmental conservation it would be possible to develop quantitative post-project evaluations of environmental policies. This could

provide valuable information to public administrators for designing and revising environmental plans and policies [6].

Another strong feature of the technique is that the assessment results of environmental efficiency can be represented visually for every municipality using spatially explicit graphs (e.g., Figure 7). Such visualizations across all Japanese municipalities can convey easily research findings to now-experts and spot easily hotspot areas or the relative performance of any given municipality. Moreover, it is possible to estimate time series for environmental efficiency for every municipality, possibly leading to the development of a new system for conceptualizing the development history of each municipality. Such applications could be useful for national government agencies to understand broader patterns nationally.

However, despite the strengths, and possibly the high relevance, of this technique for local authorities there are certain constraints that might curtail its uptake and applicability. One major constraint is the data intensiveness of the technique, which is a common characteristic of most multi-impact LCA techniques [26]. For example, in this study, the assessment data were obtained from statistical information collected by the Japanese government. However, not all of the necessary inventory datasets are currently available at the municipal level in Japan. In order to be able to apply the technique with the currently available data, we needed to simplify the estimation method. However, such simplifications might increase the uncertainty of the results [45], reducing thus their usefulness for effective decision-making at the local and the national level. For example, the inventory of industrial waste was reluctantly excluded from this assessment because the relevant data is not available at the prefectural level or by type of business. In most cases, the administrative divisions where industries operate are not the same as the divisions where they dispose their waste (particularly for industrial waste).

To increase the accuracy of studies as the one presented in this paper there is a need for complete datasets, and if possible, the development of assessment tools based on the principle of territorial benefit (Section 2.3). However, this would require the closer collaboration between local authorities and various statistics agencies in Japan in order to both obtain the necessary data that is not available at the municipal level, as well as to identify what statistical information is missing and how to collect it (e.g., request the private sector to conduct more accurate assessments and report accordingly the datasets).

Moreover, it is necessary to compare the results of method described in this study, with the results of other similar methods. This could help understand better the performance and applicability of this method for comprehensively measuring environmental loads. Several other methods could be used to measure environmental loads in different spatial scales, such as the carbon and land footprints described in Section 1 [20–25]. Such comparative analyses could identify the advantages and disadvantages of each method, as well as synthesize them in novel combinations to construct more suitable assessment theories for practical application.

Finally, to the extent possible the assessment process described in this study should be simplified, without losing its accuracy and explanatory power, so that it may be more widely utilized. Ideally, the assessment process should be also automatized to the extent possible. If tools based on this technique can update the inventory data each year with minimum time, money, and human resource investment, then they might become more appealing to local governments that lack such resources.

*4.2. Expansion of LCIA-Based Methods at the Global Scale*

This study leveraged the framework and assessment theory of LIME2 (Section 2.2). Even though there are other LCIA-based methods such as Extern E [18] and EPS [19] that are also able to calculate different environmental impacts through a single indicator, there are limited applications at the scales of municipalities and local governments. Future research should conduct such assessments for administrative divisions in other countries following LCIA-based methods that reflect local climatic conditions and use data from local statistical agencies. Moreover, it may be possible to quantify the

environmental efficiency of administrative divisions around the world through the application of a global-scale LCIA method, such as IMPACT World+ [47] and LC-IMPACT [48]. In so doing, information useful to local governments can be more widely disseminated in different parts of the world.

Such studies can provide useful information for several aspects pertaining sustainable urban development, which has become a common global policy goal, as highlighted by the adoption of SDG11 (Sustainable Cities and Communities) [49]. For example, such assessment methods could be useful for measuring local government efforts toward achieving SDG3 (Good Health and Well-being), SDG12 (Responsible Consumption and Production), SDG13 (Climate Change), and SDG15 (Life on Land). It may also contribute to the achievement of SDG17 (Partnership for the Goals) by promoting the sharing of information between local governments around the world about assessment results based on a common method.

## 5. Conclusions

This study assessed the environmental efficiency of Japanese municipalities by using LIME2, an endpoint-type LCIA method. Based on this analysis we estimate that the total damage amount of environmental loads for Japan was USD 76.6 billion (JPY 8.53 trillion) in 2015. The six largest impact categories in terms of damage amount are global warming, land use, domestic waste, biological toxicity, atmospheric pollution, and resource consumption. The damage amount per capita tends to be lower in densely populated areas, such as parts of major metropolitan areas. This suggests that environmental efficiency may be higher in more densely populated areas, particularly from the perspective of residential land use, which further suggests the importance of appropriate urban planning and development. The environmental efficiency tended to be higher in areas where industries produce large benefits relative to their environmental loads (e.g., areas with automotive and electronics industries).

Of the 21 cities designated by the government of Japan (including a special ward of Tokyo), seven ranked in the top 20% and 16 ranked in the top 50% among all municipalities in terms of environmental efficiency. This suggests that environmental efficiency is comparatively higher in many of the municipalities located in large urban areas from the perspective of productivity. Despite the ability to estimate and visualize the environmental performance of different municipalities in a comparative manner, this method (and similar methods in other parts of the world) are rarely used by local governments for environmental accounting. Future research should aim to further simplify and automate such methods in order to be more readily utilized by local governments for environmental accounting.

**Supplementary Materials:** The following are available online at http://www.mdpi.com/2071-1050/11/15/4045/s1, Figure S1: Total damage amounts for Japan by impact category (in JPY), Figure S2: Breakdowns of inventory data for total damage amount by major impact categories (in JPY), Figure S3: Spatially explicit total damage amount in Japanese municipalities (in JPY), Figure S4: Gross regional product (GRP) per capita and environmental efficiency for Japanese municipalities (in JPY), Figure S5: Assessment results for 21 major municipalities in Japan (in JPY), Table S1: Damage amounts per unit area and per capita (in JPY), Table S2: Assessment results for 21 major municipalities in Japan (in JPY).

**Author Contributions:** Conceptualization, J.Y.; Methodology, J.Y. and N.I.; Software, J.Y.; Validation, J.Y.; Formal Analysis, J.Y.; Investigation, J.Y.; Resources, J.Y.; Data Curation, J.Y.; Writing—Original Draft Preparation, J.Y.; Writing—Review and Editing, J.Y.; Visualization, J.Y.; Supervision, J.Y.; T.I.; Project Administration, J.Y.

**Funding:** This research received no external funding.

**Conflicts of Interest:** The authors declare no conflict of interest.

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
