# Peer review of "Eco-Efficiency Assessment of Japanese Municipalities Based on Environmental Impacts and Gross Regional Product"

_sustainability, doi:10.3390/su11154045_

Round 1

Reviewer 1 Report

The authors present a paper about the application of the LIME2 environmental indicator to municipalities in Japan, with a comprehensive analysis of results. In general, the research content of the article has a high quality and potential interest. However, in my opinion it needs major revision in some aspects. The following comments should be considered:

Abstract:

- Lines 18-21: I would suggest to remove this definition of the indicator from the abstract.

- Line 25: "as calculated above" should be removed.

Introduction:

- Lines 55-58: Please review this sentence, since it is not clear.

- Line 63: "there is an idea of..." should be removed.

Assessment method:

- No reference is cited in this entire section, despite it is based on the use of LIME2, developed by one of the authors.

- The authors repeat the sentence "Further details are described below". This is considered not necessary. Instead, they should explain their decisions here, such as why data from 2015 was chosen.

- Line 114: It should say "statistics are available" (remove "as").

- Line 117: I think it should say "it excludes houlhold statistics" (remove "of").

- Lines 129-138: This paragraph should be better explained. The integration factors used to translate results into monetary values are not well explained. How are they established?

- Lines 156-157: The easiness to understand this indicator is stated twice. This kind of repetition happens some other times in the manuscript. The authors should review the entire text looking for sentences like this one.

- Figure 1: Despite the indicator's framework considers inventory, impact categories, category endpoints, and safeguard subjects, only inventory and impact categories are mentioned in the assessment and discussion of results, while endpoints and safeguard subjects become completely ignored. This drives me to think if these two steps have a real value within the LIME2 indicator, or if maybe impact categories could be directly translated into Eco-index Yens.

- In section 2.3, the authors discuss the concepts of territorial occurrence and territorial consumption, but the topic of environmental responsibility has been discussed before by other authors. Again, there is no reference, and it appears the authors keep making statements of their own thoughts without searching for references and discussing their findings. Also the easy option is finally chosen, then why dedicate a whole subsection to this topic? This is usually a bad option for countries with high rates of imports, since it appears they do not cause too much environmental impact, while they are great consumers. The authors should try to improve the entire section 2, maybe taking into account some of the following articles:

Marques, A., J. Rodrigues, M. Lenzen y T. Domingos: Income-based environmental responsibility. Ecological Economics, 84:57-65, 2012.

Bastianoni, S., F.M. Pulselli y E. Tiezzi: The problem of assigning responsibility for greenhouse gas emissions. Ecological Economics, 49(3):253-257, 2004.

Cadarso, M._A., L.A. L_opez, N. G_omez y M._A. Tobarra: International trade and shared environmental responsibility by sector. An application to the Spanish economy. Ecological Economics, 83:221-235, 2012.

Munksgaard, J. y K. Pedersen: CO2 accounts for open economies: producer or consumer responsibility? Energy Policy, 29:327-334, 2001.

Rodrigues, J., T. Domingos, S. Giljum y F. Schneider: Designing an indicator of environmental responsibility. Ecological Economics, 59:256-266, 2006.

Proops, J.L.R., G. Atkinson, B.F.V. Schlotheim y S. Simon: International trade and the sustainability footprint: a practical criterion for its assessment. Ecological Economics, 28:75-97, 1999.

Lenzen, M., J. Murray, F. Sack y T. Wiedmann: Shared producer and consumer responsibility - theory and practice. Ecological Economics, 61(1):27-42, 2007.

Laroche, M., J. Bergeron y G. Barbaro-Forleo: Targeting consumers who are willing to pay more for environmentally friendly products. Journal of Consumer Marketing, 18(6):503-520, 2001.

Ferng, J.J.: Allocating the responsibility of CO2 over-emissions from the perspectives of bene_t principle and ecological de_cit. Ecological Economics, 46:121-141, 2003.

Andrew, R. y V. Forgie: A three-perspective view of greenhouse gas emission responsibilities in New Zealand. Ecological Economics, 68:194-204, 2008.

-        Line 198: it should say “LIME2”

-        Lines 257-258: There should be a reference here.

-        Lines 259-261. This has already been said before.

-        Lines 286-287: Why the construction process is taken into account for indoor air pollution?

Section 3:

-        Figure 4: Maybe a comparative bar plot would fit better for this purpose.

-        Lines 326-328: The authors repeat the same statement twice in a single sentence.

-        Lines 360-371: format.

-        Figure 6: some texts appear behind the drawings.

-        Line 419: This sentence is not necessary.

-        Line 420: GRP has not been defined before.

-        Line 462: This sentence is repetitive.

Section 4: The conclusions should be considerably improved. In the current state of this section, there is a summary of the article and future challenges, as well as the repetition of some findings already included in the results section, but no conclusions appear to be drawn from the study in general.

Author Response

Thank you very much for your many valuable comments. We indicated the revised points in red in the PDF document as the responses to your comments.

Reviewer 2 Report

The paper is well written and touches an interesting subject.

As a whole, the manuscript contents are well presented. However, the paper requires some correction. My comments and suggestions are as follows:

-The contributions of the study should be highlighted further and research gaps could be discussed more which will consequently lead to research questions which are not properly expressed.

- The first sections should include more references. Overall, the list of references includes more web sites and reports for the case studies which is understandable, however, research of high level on similar or ideally same subject should be presented.

-Conclusions should provide more in-depth information and implications should be recorded.

-In terms of English, some minor revisions are required to improve clarity and overall reading.

Author Response

(The authors gave the same response as above.)

Reviewer 3 Report

The paper is provided about a very strong and comprehensive coverage of measuring and assessing the environmental impact in the Japan. 

There is no doubt it can be usedas one of the best tools in the Japan but there are some concernes about the LCIA that in summary are:

- how this tools deals with the Exports and Impotrs?

- when we are talking about LCIA, labour, the impacts happended in the origin of the materials how can be considered? And how about the export of the materials and products are going to be used in the othere countries can be decided and measured?

- To measure LCIA in the japan it can work very well

- I don't know any solution for this issue 

Author Response

(The authors gave the same response as above.)

Round 2

Reviewer 1 Report

The authors have attended the suggestions made by the reviewers. Apart from some typos that the editors should revise, I think the manuscript is ready for publication.

Author Response

We apologize for the delayed submission of the revised version.

Thank you very much for doing a thorough editing to our paper. We accepted most of it. And the corrected part is shown in red.